# Why not to pick your nose: Association between nose picking and SARS-CoV-2 incidence, a cohort study in hospital health care workers

**A. H. Ayesha Lavell**[1,2], **Joeri Tijdink**[3,4], **David T. P. Buis**[1,2], **Yvo M. Smulders**[1,2], **Marije K. Bomers**[1,2], **Jonne J. Sikkens**[1,2]*

**1** Department of Internal Medicine, Amsterdam UMC location Vrije Universiteit Amsterdam, Amsterdam, The Netherlands, **2** Amsterdam Institute for Infection and Immunity, Amsterdam, The Netherlands, **3** Department Ethics, Law and Humanities, Amsterdam UMC location Vrije Universiteit Amsterdam, Amsterdam, The Netherlands, **4** Department of Philosophy, Vrije Universiteit Amsterdam, Amsterdam, The Netherlands

* j.sikkens@amsterdamumc.nl

**Data Availability Statement:** Data are all contained within the paper and Supporting Information files.

## Abstract

### Background

Hospital health care workers (HCW) are at increased risk of contracting SARS-CoV-2. We investigated whether certain behavioral and physical features, e.g. nose picking and wearing glasses, are associated with infection risk.

### Aim

To assess the association between nose picking and related behavioral or physical features (nail biting, wearing glasses, and having a beard) and the incidence of SARS-CoV-2-infection.

### Methods

In a cohort study among 404 HCW in two university medical centers in the Netherlands, SARS-CoV-2-specific antibodies were prospectively measured during the first phase of the pandemic. For this study HCW received an additional retrospective survey regarding behavioral (e.g. nose picking) and physical features.

### Results

In total 219 HCW completed the survey (response rate 52%), and 34/219 (15.5%) became SARS-CoV-2 seropositive during follow-up from March 2020 till October 2020. The majority of HCW (185/219, 84.5%) reported picking their nose at least incidentally, with frequency varying between monthly, weekly and daily. SARS-CoV-2 incidence was higher in nose picking HCW compared to participants who refrained from nose picking (32/185: 17.3% vs. 2/34: 5.9%, OR 3.80, 95% CI 1.05 to 24.52), adjusted for exposure to COVID-19. No

**Funding:** This work was funded by the Netherlands Organization for Health Research and Development ZonMw (S3 study, grant agreement no. 10430022010023 to M.K.B.) and the Corona Research Fund Amsterdam UMC. The funders were not involved in designing the study, data collection, analysis or interpreting the data, writing the manuscript or the decision to submit the article for publication.

**Competing interests:** The authors have declared that no competing interests exist.

association was observed between nail biting, wearing glasses, or having a beard, and the incidence of SARS-CoV-2 infection.

## Conclusion

Nose picking among HCW is associated with an increased risk of contracting a SARS-CoV-2 infection. We therefore recommend health care facilities to create more awareness, e.g. by educational sessions or implementing recommendations against nose picking in infection prevention guidelines.

## Introduction

The end of 2019 marked the beginning of the global SARS-CoV-2 pandemic [1]. Worldwide, preventive measurements were introduced aiming to reduce physical contact, and droplet and aerosol transmission [2, 3]. In healthcare facilities, guidelines recommended the use of personal protective equipment (PPE) for those working in direct patient care, including the use of face-masks, a gown and goggles/face shields, as well as gloves and strict hand hygiene protocols [3–5].

Despite these guidelines, health care workers (HCW) are more likely to contract SARS-CoV-2 infection (hazard ratio [HR] 3.92, 95% CI 1.79 to 8.62, for those working with COVID-19 patients compared with HCW not working in patient care [6]), with risk factors including sub-optimal hand hygiene and use of PPE [7, 8]. Although the main route of inoculation is via respiratory mucosa [2, 9], it is unknown whether transmission of SARS-CoV-2 is affected by habitual hand-mucosa contact, as occurs in nose picking and nail biting. Studies suggest that a large proportion of the adult population regularly pick their nose [10, 11]. It can be hypothesized that regular nose picking and nail biting in an environment with high levels of circulating virus enables the virus's transfer to the nasal or oral mucosa, as is seen with nose picking and *S. aureus* nasal carriage [12]. Infection risk may be further increased when mucosa are damaged, e.g. from the strain due to repetitive nasal finger penetration [13]. Also, it is unknown whether transmission risk is affected by physical features influencing fit of PPE and susceptibility to droplets, for example having a beard or wearing glasses [14, 15]. The latter is of interest since the ocular mucosa has been identified as another possible route of SARS-CoV-2 transmission [16, 17].

In this study, we aim to examine the effects of an individual's behavioral and physical features on SARS-CoV-2 transmission in a prospective cohort study of HCW. More precisely, we are interested in the role of skin and mucosa manipulating-related behavior (nose picking and nail biting), and physical features influencing fit of PPE (having a beard), and susceptibility to droplets (wearing glasses).

## Methods

### Study design and participants

The S3 cohort was set up in March 2020 to study incidence of SARS-CoV-2 infection and identify its potential risk factors among HCW working in the Amsterdam University Medical Centers (S3 cohort; NL 73478.029.20, Netherlands Trial Register NL8645). SARS-CoV-2 incidence was monitored by both serologic surveillance and self-reported nucleic acid amplification tests (NAAT)-results; risk factors for infection were identified by means of surveys regarding work- and community-related COVID-19 exposure. A more comprehensive overview of the original cohort and results of this study is described elsewhere [6]. The current

study is part of a S3 cohort sub study on behavioral features and mental health, for which HCW already participating in the main cohort were asked to participate in October 2020. Of a total of 801 HCW, 404 (50.4%) agreed to participate in the sub study for which written informed consent was obtained. These 404 received an additional online retrospective survey in 2021 on behavioral and physical features that may influence infection rates, i.e. practices as nose picking, nail biting, wearing glasses or having a beard (S1 File). All survey data was collected using Castor Electronic Data Capture (EDC) [18]. The study was reported as per STROBE guidelines and approved by the medical ethics review committee of both hospitals of the Amsterdam University Medical Centers.

## COVID-19 exposure

'Working in COVID-19 patient care' and 'contact with a community member or coworker with COVID-19' were associated with an increased risk of contracting SARS-CoV-2 infection in the original S3 study cohort [6]. To be able to adjust for possible confounding of these factors, participants were categorized as either 'working in COVID-19 patient care', 'working in non-COVID-19 patient care', or 'not working in patient care'. Additionally, participants that reported close contact with a symptomatic coworker or community member were identified. Besides NAAT-confirmed SARS-CoV-2-infected contacts, community members with high suspicion of infection were also included before NAAT-testing became widely available (March to June 2020) [19].

## Outcome measures

**SARS-CoV-2 infection.**   SARS-CoV-2 infection was defined as a self-reported positive NAAT result and/or presence of SARS-CoV-2-specific antibodies, as detected by measuring total-Ig against S1-RBD using the commercially available Wantai enzyme-linked immunosorbent assay (ELISA) [20]. The Wantai ELISA has a specificity of 99.6% and sensitivity of $\geq$95.4% in mild or asymptomatic cases 14 days after the onset of illness [21–23].

## Infection control practices

During this study both hospitals instituted identical infection control measures. In-hospital social distancing included keeping 1.5m distance between individuals not wearing PPE, working from home for non-essential personnel, etc. Dedicated COVID-19 wards and intensive care units were established; HCW working with possible or proven COVID-19 patients wore PPE consisting of gloves, gowns, goggles and IIR surgical masks during non-aerosol generating care, or FFP2 masks combined with a cap during high-risk aerosol generating procedures and on the intensive care ward. No PPE was recommended outside COVID-19 patient care. From October 1, 2020, onwards all personnel were requested to wear a face shield or face mask in public spaces in accordance with national guidelines. A more in detail description was provided previously [6].

## Statistical analysis

The associations between the incidence of SARS-CoV-2 infections (dependent variable) and nose picking, nail biting, wearing glasses, and having a beard (independent variables), were assessed by logistic regression analysis. The multivariable model was adjusted for determinants previously shown to be associated with COVID-19 incidence in the cohort: working in direct COVID-19 patient care, and contact with coworkers or community members with SARS-CoV-2 infection [6]. Mixed model logistic regression was performed as sensitivity analysis to

adjust for possible clustering within hospitals and specific departments. We did not perform survival analysis since the proportional hazards assumption did not hold due to the highly variable COVID-19 incidence and changing PPE guidelines over time. Data analysis was performed using R version 4.0.3. Significance level was set at an alpha of 5%, defined as a 95% confidence interval (CI) excluding an odds ratio of 1. Graphs were made using GraphpadPrism version 9.1.0.

## Results

### Descriptives

A total of 219 HCWs (52.4% of 404 participants) completed the survey regarding behavioral and physical features of interest. Habitual nose picking, varying from monthly, weekly to daily, was disclosed by 185 (85%) respondents. Nose pickers were younger than non-nose pickers (median age in years for nose pickers 44 (IQR 36 to 56) and for non-nose pickers 53 (IQR 46 to 57), and males reported more frequently nose picking (90%) compared with females (83%). Doctors were the most frequent nose pickers (residents: 100% and specialists: 91%), followed by support staff (86%) and nurses (80%). Nail biting (monthly, weekly, daily or hourly) was less frequently reported (33%), 158 (72%) participants reported wearing glasses and 18/52 males (35%) reported having a beard (Table 1). By October 2020, which was the start of the second pandemic wave in the Netherlands, 34/219 (16%) HCW were SARS-CoV-2 seropositive. Only 2 (6%) of these seropositive participants never picked their nose, while 9 (27%) reported monthly nose picking, 12 (35%) weekly and 11 (32%) daily. None of the participants reported to pick their nose every hour (S1 Table and S1 Fig).

### Role of behavior manipulating skin and mucosa and the risk of COVID-19: Nose picking and nail biting

We found an increased risk of SARS-CoV-2 infection for nose pickers (COVID-19 rate 32/185: 17.3%, Table 2) compared to participants who refrained from nose picking (2/34: 5.9%, OR 3.80, 95% CI 1.05 to 24.52, Table 2), adjusted for exposure to COVID-19. When divided into subgroups based on nose picking frequency, all nose picking groups had higher infection rates compared to the non-nose picking group, but the only significant subgroup difference was between those reporting weekly nose picking and those who never picked their nose (subgroup analysis shown in Fig 1). Post-hoc sensitivity analysis, to adjust for possible clustering within workplace showed the same trend (OR 3.74, 95% CI 0.98–25.06, S2 Table). We did not

**Table 1. Demographics of behavioral and physical features.**

| Response | Sex | | Age in years, median (IQR) | | Job title | | | | Overall (n = 219) |
|---|---|---|---|---|---|---|---|---|---|
| | Male n = 52 (23.9%) | Female n = 166 (76.1%) | Behavior present | Behavior absent | Nurse n = 99 (45.2%) | Resident n = 10 (4.6%) | Specialist n = 33 (15.1%) | Support staff n = 77 (35.2%) | - |
| Nose picking, n (%) | 47 (90.4%) | 138 (83.1%) | 44 (36–56) | 53 (46–57) | 79 (79.8%) | 10 (100%) | 30 (90.9%) | 66 (85.7%) | 185 (84.5%) |
| Nail biting, n (%) | 19 (36.5%) | 53 (31.9%) | 41 (33–54) | 49 (38–57) | 35 (35.4%) | 6 (60.0%) | 7 (21.2%) | 24 (31.2%) | 72 (32.9%) |
| Glasses, n (%) | 32 (61.5%) | 113 (68.1%) | 53 (42–58) | 38 (30–45) | 64 (64.6%) | 3 (30.0%) | 20 (60.6%) | 59 (76.6%) | 146 (66.7%) |
| Beard, n (%) | 16 (30.8%) | - | 43 (37–53) | 46 (38–57) | 6/12 (50.0%) | 1/3 (33.3%) | 1/17 (5.9%) | 8/20 (40.0%) | - |

Baseline characteristics of nose pickers (varying from monthly, weekly or daily), nail biters (monthly, weekly, daily or hourly), participants wearing glasses (weekly or daily) and having a beard (weekly or daily).

**Table 2. Association between nose picking, nail biting, wearing glasses and having a beard, and the incidence of SARS-CoV-2.**

| Behaviors (dichotomous) | Event rate | | Annualized event rate | | Crude OR (95% CI) | Adjusted OR (95% CI) |
|---|---|---|---|---|---|---|
| | Behavior present | Behavior absent | Behavior present | Behavior absent | | |
| Nose picking | 32/185 (17.3%) | 2/34 (5.9%) | 29.7% | 10.1% | 3.35 (0.95 to 21.30) | **3.80 (1.05 to 24.52)** |
| Nail biting | 10/72 (13.9%) | 24/147 (16.3%) | 23.8% | 28.0% | 0.83 (0.36 to 1.79) | **0.97 (0.41 to 2.18)** |
| Glasses | 17/146 (11.6%) | 17/73 (23.3%) | 20.0% | 40.0% | 0.43 (0.21 to 0.91) | **0.49 (0.23 to 1.06)** |
| Beard* | 2/16 (12.5%) | 4/36 (11.1%) | 21.4% | 19.0% | 1.14 (0.15 to 6.59) | **1.06 (0.13 to 6.80)** |

Results of uni- and multivariable logistic regression assessing the association between behavioral and physical features and the outcome SARS-CoV-2 infection (event). Nose picking and nail biting were dichotomized into never versus yes (variating from monthly, weekly, daily or every hour). Wearing glasses or having a beard were dichotomized into never or monthly versus weekly or daily.

* Males only. Annualized event rate was calculated by dividing the event rate by 7 (approximately the months of study follow-up), multiplying by 12 and converted into percentage of total participants according to behavior.

Models were adjusted for possible confounding of working in COVID-19 patient care and contact with a COVID-19 infected coworker (NAAT [nucleic acid amplification test] confirmed) or community member (between March and June 2020 suspected and NAAT confirmed, between June and October 2020 NAAT confirmed).

find a significant association between nail biting and the incidence of SARS-CoV-2 infection (COVID-19 rate in nail biters 10/72: 13.9%, versus non-nail biters 24/147: 16.3%, OR 0.97, 95% CI 0.41 to 2.18, Table 2), adjusted for exposure to COVID-19.

## Physical features affecting fit of PPE and risk of COVID -19: Wearing glasses and having a beard

Glasses-wearing respondents contracted SARS-CoV-2 infection at a lower rate compared to those without glasses, but the difference was not significant in the adjusted model (17/146: 11,6% versus 17/73: 23.3%; OR 0.49, 95% CI 0.23 to 1.06, Table 2). We did not find an association between having a beard and the incidence of COVID-19 in men (2/16: 12.5% versus 4/36: 11.1%; OR 1.06, 95% CI 0.13 to 6.80, Table 2).

## Discussion

In this hospital HCW cohort study we found that nose picking is associated with an increased incidence of SARS-CoV-2 infection.

Nose picking has not been reported before as a risk factor for contracting SARS-CoV-2. Our findings highlight the importance of the nasal cavity as a main transit port for SARS-CoV-2 [2, 9, 24]. Nose picking may facilitate viral entry by directly introducing virus particles present on the hands to the nose, thus facilitating infection [13, 25, 26]. The viral load in the nasal mucosa is high in the days after contracting a SARS-CoV-2 infection, even before the onset of symptoms and in patients that remain asymptomatic [24, 27]. Subsequently, nose picking HCW who are infected with SARS-CoV-2 could contaminate the work environment, potentially leading to further transmission [28, 29]. SARS-CoV-2 transmission from HCW-to-HCW is an important problem in hospitals [6], perhaps the role of nose picking is underestimated in this regard.

We did not find an association between nail biting and the incidence of SARS-CoV-2 infection. This might be explained by the protective effects of salivary proteins which were recently demonstrated to inhibit SARS-CoV-2 spike protein binding to the ACE2 receptor [30], making the mouth merely an exit rather than an entrance route for viral transmission [24].

An important strength of our study is the prospective longitudinal serological sampling from the start of the very first phase of the pandemic. The well-characterized cohort allowed

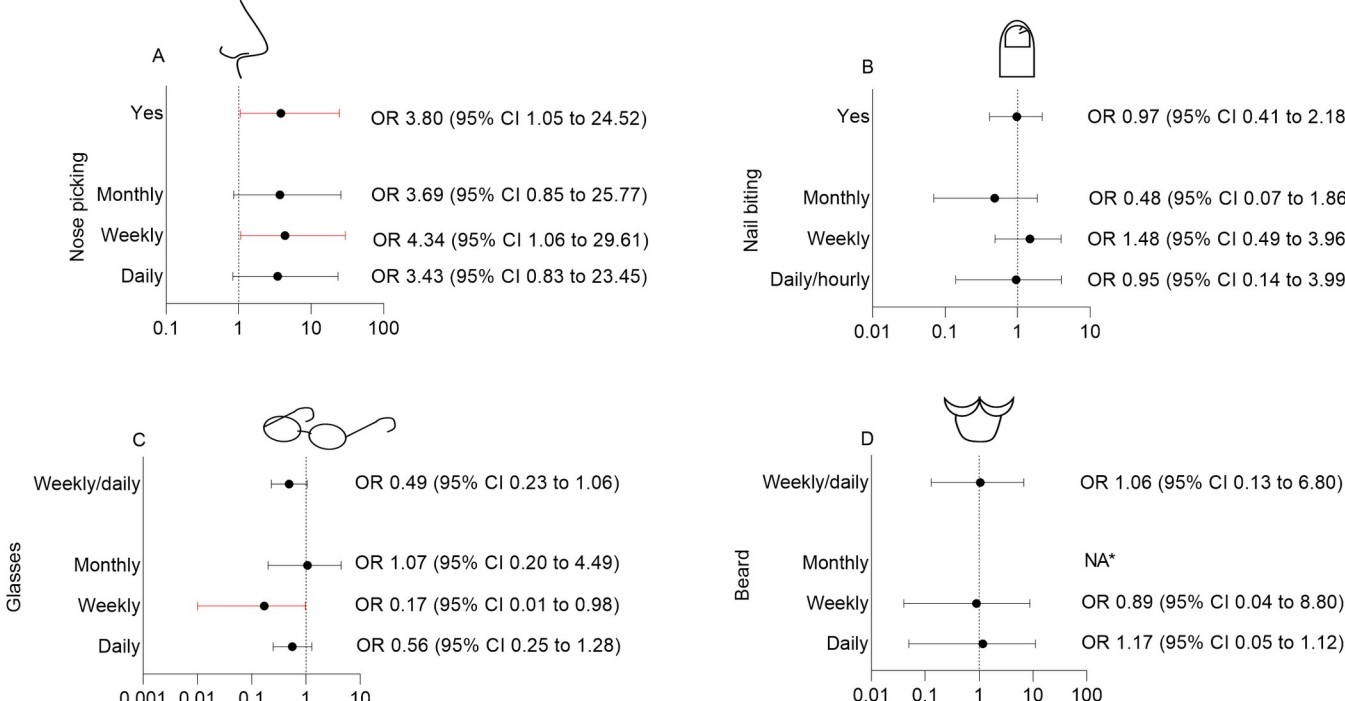

**Fig 1. ORs and 95% CI of the association between nose picking, nail biting, wearing glasses or having a beard, and the incidence of SARS-CoV-2.** Plots showing Odds ratios (OR) and confidence intervals (CI) of the association between nose picking (A), nail biting (B), wearing glasses (C), having a beard (D) and SARS-CoV-2 infection, assessed by multivariable logistic regression analysis corrected for exposure to COVID-19 (i.e. working with COVID-19 patients and contact with a coworker or community-contact with SARS-CoV-2 infection). The association between nose picking, nail biting, wearing glasses and having a beard as ordinal determinants and the outcome SARS-CoV-2 infection is depicted (any frequency versus never), as well as dichotomized determinants: no (never) versus yes (monthly/weekly/daily/hourly) for nose picking and nail biting and no (never/monthly) versus yes (weekly/daily) for wearing glasses and having a beard. Significant 95% CI's are indicated by red error bars. *NA due to non-positivity.

for the adjustment for relevant confounding factors. In addition, we were able to include a population of HCW (nurses, doctors, support staff) with characteristics generalizable to the Dutch population of HCW [31]. Finally, the subject of facial and skin-related behavior and specifically nose picking has not been studied with regard to infection transmission of COVID-19.

Some limitations need to be discussed. We found a relevant, but not statistically significant difference in SARS-CoV-2 infection incidence between HCW wearing glasses and those that did not, which may have been caused by an insufficient sample size. The time-interval between the serology measurements (March-October 2020) and the survey exploring behavioral and physical features (December 2021) could have introduced recall bias and potential shifting of (nose picking) behavior [32, 33]. Also, we did not ask whether HCW committed to nose picking and nail biting when on the work floor, or the specifics of inter variability between nose pickers, e.g. the depth of penetration and eating of boogers. The current study was performed in the pre-Omicron and pre-vaccination era, so implications for current practice could be influenced by changing viral characteristics such as virus-specific transmission dynamics and by differences in host immunity [34]. However, identifying and addressing readily preventable sources of transmission remains important to limit in-hospital spread of SARS-CoV-2 and (probably) other respiratory viruses, both to patients and co-workers, in any epidemic. For this reason, we feel our findings are relevant despite these limitations and underline the importance of preventive measures and proper hand hygiene when working in healthcare.

## Conclusion

In conclusion, this is the first study that shows that nose picking by HCW is associated with an increased risk of contracting COVID-19. It is surprising to observe the extensiveness in which the scientific community (including our own study team) has researched all sorts of SARS-CoV-2 transmission routes, risk factors and protective measures; yet assessing the role of simple behavioral and physical properties has so far been overlooked. Possibly this sensitive subject is still taboo in the health care profession. It is commendable we assume HCWs to not portray bad habits, yet we too are only human after all, as illustrated by the pivotal proportion of nose pickers in our cohort (84.5%). Considering guideline recommendations include e.g. illustrations of appropriate masks for those with facial hair despite the lack of any real-life evidence [35], nose picking deserves more consideration as a potential health hazard, and explicit recommendations against nose picking should be included in the same SARS-CoV-2 infection prevention guidelines. Future research could examine the effectiveness of interventions addressing behavior (like awareness campaigns, or the use of nail polish with an unpleasant smell) [36, 37], treating the underlying cause of nose picking (e.g. by using saline spray to reduce mucus [38, 39]) or using nasal disinfectant spray in SARS-CoV-2 infected individuals to counteract viral shedding [40–42].

## Supporting information

**S1 File. Survey regarding behavioral and physical features.**
(DOCX)

**S1 Fig. Proportion of behavioral and physical features.** A-D. Bar charts showing the proportion of nose picking, nail biting, and wearing glasses or having a beard in SARS-CoV-2 seropositive and seronegative participants.
(TIF)

**S1 Table. Baseline characteristics demographic, behavioral and physical features in relation to SARS-CoV-2 status.** Baseline characteristics of participants (n = 219) divided into SARS-CoV-2 seropositive and seronegative subgroups. *Males only.
(DOCX)

**S2 Table. Sensitivity analysis: Association between nose picking, nail biting, wearing glasses or having a beard, and the incidence of SARS-CoV-2 adjusted for working in different departments.** Results of mixed model logistic regression assessing the association between behaviors or physical features and the outcome SARS-CoV-2 infection. Nose picking and nail biting were dichotomized into never versus yes (variating from monthly, weekly, daily to every hour). Wearing glasses or having a beard were dichotomized into no (never or monthly) versus yes (variating from weekly to daily) * Males only. Models were corrected for possible confounding of working in different hospitals (location AMC or location VUmc) and departments (intensive care unit (ICU), emergency department (ED), nursing ward, non-COVID-19 patient care, non-patient care) and contact with a COVID-19 infected coworker of community member.
(DOCX)

**S3 Table. De-identified dataset.** SARS-CoV-2 is coded as 0 for seronegative and 1 for seropositive; sex as 0 for male and 1 for female; exposed as 0 for not exposed and 1 for exposed (working in direct care for COVID-19 patients); community contact and coworker contact as 0 for no and 1 for yes; nose picking, nail biting, wearing glasses and having a beard as 0 for

never, 1 for monthly, 2 for weekly, 3 for daily and 4 for every hour.
(XLSX)

## Acknowledgments

We thank all participating healthcare workers of Amsterdam UMC, who took time to facilitate this study in the midst of the pandemic, for their contribution.

## Author Contributions

**Conceptualization:** Joeri Tijdink, Marije K. Bomers, Jonne J. Sikkens.

**Data curation:** A. H. Ayesha Lavell, David T. P. Buis.

**Formal analysis:** A. H. Ayesha Lavell, David T. P. Buis, Jonne J. Sikkens.

**Funding acquisition:** Marije K. Bomers, Jonne J. Sikkens.

**Investigation:** A. H. Ayesha Lavell, Joeri Tijdink, Marije K. Bomers, Jonne J. Sikkens.

**Methodology:** A. H. Ayesha Lavell, Joeri Tijdink, Marije K. Bomers, Jonne J. Sikkens.

**Project administration:** A. H. Ayesha Lavell, David T. P. Buis.

**Supervision:** Joeri Tijdink, Yvo M. Smulders, Marije K. Bomers, Jonne J. Sikkens.

**Validation:** A. H. Ayesha Lavell, Marije K. Bomers, Jonne J. Sikkens.

**Visualization:** A. H. Ayesha Lavell, Marije K. Bomers, Jonne J. Sikkens.

**Writing – original draft:** A. H. Ayesha Lavell, Joeri Tijdink, David T. P. Buis, Yvo M. Smulders, Marije K. Bomers, Jonne J. Sikkens.

**Writing – review & editing:** A. H. Ayesha Lavell, Joeri Tijdink, David T. P. Buis, Yvo M. Smulders, Marije K. Bomers, Jonne J. Sikkens.

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
