## [Decision Letter · Decision Letter 0]

20 Apr 2023

PONE-D-23-03063Why not to pick your nose: association between nose-picking and SARS-CoV-2 incidence, a cohort study in hospital health care workersPLOS ONE

Dear Dr. Ayesha,

Thank you for submitting your manuscript to PLOS ONE. After careful consideration, we feel that it has merit but does not fully meet PLOS ONE’s publication criteria as it currently stands. Therefore, we invite you to submit a revised version of the manuscript that addresses the points raised during the review process.

We look forward to receiving your revised manuscript.

Kind regards,

Tope Michael Ipinnimo

Academic Editor

PLOS ONE

Journal Requirements:

a) The name of the colleague or the details of the professional service that edited your manuscript.

b) A copy of your manuscript showing your changes by either highlighting them or using track changes (uploaded as a *supporting information* file).

c) A clean copy of the edited manuscript (uploaded as the new *manuscript* file).

3. Please ensure that you have specified (1) whether consent was informed and (2) what type you obtained (for instance, written or verbal, and if verbal, how it was documented and witnessed). If your study included minors, state whether you obtained consent from parents or guardians. If the need for consent was waived by the ethics committee, please include this information.

"None of the authors declare any competing interests."

6. We note that you have indicated that data from this study are available upon request. PLOS only allows data to be available upon request if there are legal or ethical restrictions on sharing data publicly. For more information on unacceptable data access restrictions, please see http://journals.plos.org/plosone/s/data-availability#loc-unacceptable-data-access-restrictions. 

**Additional Editor Comments:**

1. Please kindly proof read the work for typographical and grammatical errors

2. Is this really a prospective cohort study if the data collection regarding behavioral and facial characteristics were collected retrospectively?

3. "Participants were categorized into working in COVID-19 patient care and working in non-COVID-19 patient care or not working in patient care" (under the methods: COVID-19 Exposure - Line 109 and 110). Why did the authors categorize the cohort into these groups? The outcome measure is SARS-CoV-2 infection and the exposure/independent variables were nose-picking, nail-biting etc

Reviewers' comments:

Reviewer's Responses to Questions

**Comments to the Author**

1. Is the manuscript technically sound, and do the data support the conclusions?

Reviewer #1: Yes

Reviewer #2: Yes

2. Has the statistical analysis been performed appropriately and rigorously? 

Reviewer #1: Yes

Reviewer #2: Yes

3. Have the authors made all data underlying the findings in their manuscript fully available?

Reviewer #1: Yes

Reviewer #2: No

4. Is the manuscript presented in an intelligible fashion and written in standard English?

Reviewer #1: Yes

Reviewer #2: Yes

5. Review Comments to the Author

Reviewer #1: General feed- back

This study examined the risk to contract SARS-CoV-2 infection by some behavioural habits among 404 health care workers in the Netherlands surveyed online on behavioural during March 2020- October 2020, hence during the first pandemic year. SARS-CoV-2 infection was defined as a self-reported positive NAAT result and/or presence of SARS120 CoV-2-specific antibodies. Several endpoints were investigated by multivariable logistic regression: nose picking, nail-biting, wearing glasses, and having a beard. Self-reported nose picking was he only behaviour significantly associated with SARC-CoVI-2 infection [OR 3.80, 95% CI 1.05 to 57 24.52].

Specific comments

This study is interesting to be published, although some points needs to be address

First of all this study was conducted during the first pandemic year, when vaccines were not available yet and more aggressive yet less infectious viral strains were circulating. The spread of Omicron variant completely changed the scenario. Although HCWs were mostly adequately vaccinated by end of 2021, an increase of primary as well as secondary infections was observed from December 2021 on, with the diffusion of Omicron. Although generally mild, the vast majority of infections were acquired outside health care premises, suggesting that vaccination was not effective in preventing asymptomatic/mild-moderate infections during the Omicron wave. By contrast in hospital, where risk reduction measures (face mask, hand hygiene, social distancing, etc.) were observed, SARS-CoV-2 infections were low. This study reports this critical finding [PMID: 36016284]

This study used serology to confirm past SARS-CoV-2 infection. The ideal would be to swab test regularly (every 3-6 days) HCW in the nose, to detect any asymptomatic positivity.

Discussion

Lines 246: “even though the noseat size of Europeans is above world average”… this seems a n odds assertion

Conclusions (lines 248-250): “Future research could examine ….. e.g. by using saline spray to reduce mucus”. Here it is worth mentioning some relevant studies on disinfectants tested against SARS-CoV-2 in vitro as well as in vivo, to reduce the viral shedding time from patients affected by mild-moderate or asymtpomatic disease (PMID: 36676046; PMID: 36432693; PMID: 36432693)

Reviewer #2: General comments

This interesting study investigated the association between nose-picking, nail-biting, glass-wearing and beard-keeping and COVID-19 infection among healthcare workers in the Netherlands.

The study is well-conceived and conducted with sound methodology and appropriate statistical methods.

The results are presented clearly and subsequent discussion was in line with the study outcome.

Limitations were also well-highlighted.

Specific comments

Table 1 to be re-formatted to include the lateral border

6. PLOS authors have the option to publish the peer review history of their article (what does this mean?). If published, this will include your full peer review and any attached files.

Reviewer #1: No

Reviewer #2: **Yes: **Joseph O. Fadare

---

## [Author Response · Author response to Decision Letter 0]

20 Jun 2023

Point by point response to editor and reviewers

Additional Editor Comments:

1. Please kindly proof read the work for typographical and grammatical errors

Answer: A thorough proof read by the entire author team has taken place, which has resulted in changing the phrasing of several main expressions, such as wearing glasses (instead of bespectacled) and having a beard (instead of bearded). 

2. Is this really a prospective cohort study if the data collection regarding behavioral and facial characteristics were collected retrospectively?

Answer: We agree with the editor that our cohort study is a mixture of both prospectively and retrospectively collected data. Therefore we rewrote the following sentence:

“In a cohort study among 404 HCW in two university medical centers in the Netherlands, SARS-CoV-2-specific antibodies were prospectively monthly measured during the first phase of the pandemic. For this study HCW received an additional retrospective survey regarding behavioral (e.g. nose-picking) and facial characteristics.“ (Abstract, page 2, line 48-51) 

3. “Participants were categorized into working in COVID-19 patient care and working in non-COVID-19 patient care or not working in patient care” (under the methods: COVID-19 Exposure - Line 109 and 110). Why did the authors categorize the cohort into these groups? The outcome measure is SARS-CoV-2 infection and the exposure/independent variables were nose-picking, nail-biting etc

Answer: In our original publication of the S3 study [reference 6 in the main manuscript, PMID: 34319354] participants working directly with COVID-19 patients were at increased risk of contracting a SARS-CoV-2 infection during the first phase of the pandemic, compared with those working in non-COVID-19 patient care (hazard ratio [HR], 2.25; 95% CI, 1.17-4.30) and those not working in patient care (HR 3.92; 95% CI, 1.79-8.62). Therefore we used the same categories in our current study to be able to adjust for possible confounding of working in COVID-19 patient care. To clarify this, we added the following sentence to the method section: 

“Working in COVID-19 patient care and contact with a community member or coworker with COVID-19 were associated with an increased risk of contracting SARS-CoV-2 infection. 6 To be able to adjust for possible confounding of these factors participants were categorized ...” (Methods, page 4, line 111-113)

Reviewers' comments:

1. Is the manuscript technically sound, and do the data support the conclusions?

Reviewer #1: Yes

Reviewer #2: Yes

2. Has the statistical analysis been performed appropriately and rigorously? 

Reviewer #1: Yes

Reviewer #2: Yes

3. Have the authors made all data underlying the findings in their manuscript fully available?

Reviewer #1: Yes

Reviewer #2: No

Answer: We consulted the privacy officer of the Amsterdam UMC and were able to share the data set supporting our results.

4. Is the manuscript presented in an intelligible fashion and written in standard English?

Reviewer #1: Yes

Reviewer #2: Yes

5. Review Comments to the Author

Reviewer #1: General feed- back

This study examined the risk to contract SARS-CoV-2 infection by some behavioural habits among 404 health care workers in the Netherlands surveyed online on behavioural during March 2020- October 2020, hence during the first pandemic year. SARS-CoV-2 infection was defined as a self-reported positive NAAT result and/or presence of SARS120 CoV-2-specific antibodies. Several endpoints were investigated by multivariable logistic regression: nose picking, nail-biting, wearing glasses, and having a beard. Self-reported nose picking was he only behaviour significantly associated with SARS-CoV-2 infection [OR 3.80, 95% CI 1.05 to 57 24.52].

Specific comments

This study is interesting to be published, although some points needs to be address

First of all this study was conducted during the first pandemic year, when vaccines were not available yet and more aggressive yet less infectious viral strains were circulating. The spread of Omicron variant completely changed the scenario. Although HCWs were mostly adequately vaccinated by end of 2021, an increase of primary as well as secondary infections was observed from December 2021 on, with the diffusion of Omicron. Although generally mild, the vast majority of infections were acquired outside health care premises, suggesting that vaccination was not effective in preventing asymptomatic/mild-moderate infections during the Omicron wave. By contrast in hospital, where risk reduction measures (face mask, hand hygiene, social distancing, etc.) were observed, SARS-CoV-2 infections were low. This study reports this critical finding [PMID: 36016284]

Answer: We agree with the reviewer that the generalizability of our findings in the current pandemic phase with higher levels of immunity, altered strain transmissibility, and different masking habits, is unclear[PMID: 36016284]. However, identifying and addressing readily preventable sources of transmission remains important to limit in-hospital spread of SARS-CoV-2 and probably other respiratory viruses, both to patients and co-workers, in any epidemic. For this reason, we feel our findings are relevant despite these limitations and underline the importance of preventive measures and proper hand hygiene when working in healthcare. We adjusted the limitation section in the discussion correspondingly. (Discussion, page 8, line 248-254)

This study used serology to confirm past SARS-CoV-2 infection. The ideal would be to swab test regularly (every 3-6 days) HCW in the nose, to detect any asymptomatic positivity.

Answer: In the first phase of the pandemic, when we collected the data in 2020, testing by NAAT (or PCR) was not widely available yet. Therefore, due to scarcity, only symptomatic HCW were tested by using a nasopharyngeal swab. We therefore performed regular serological follow-up to be able to detect also asymptomatic individuals, additionally to self-reported results of NAATs in symptomatic individuals. The used Wantai ELISA total-Ig against S1-RBD has an specificity of 99.6% and sensitivity of 95.4% in mild or asymptomatic cases and 97.5% in severe cases >14 days after the onset of illness. (Report of Dutch Society of Medical Microbiology* and PMID: 33574119) This is close to the sensitivity (97%) of a single nasopharyngeal swab [PMID: 33857405]). Therefore we believe that the rate of false negatives in our population is low. We added the specificity and sensitivity of the Wantai essay to the methods section (Methods, page 4, line 124-126). 

* van den Beld M, Report Status of the validation of ELISA and auto-analyser antibody tests for SARS-CoV-2 diagnostics: considerations for use Status as at 15 July 2020. Avalaible: status-validation-elisa-and-auto-analysers_2020715_final.pdf (nvmm.nl)

Discussion

Lines 246: “even though the noseat size of Europeans is above world average”… this seems an odds assertion

Answer: In theory a larger nose might enhance the depth of penetration while nose picking and thus potentially increase the risk of getting infected. However, this is currently not (yet) scientifically proven we removed the assertion from our manuscript. 

Conclusions (lines 248-250): “Future research could examine ….. e.g. by using saline spray to reduce mucus”. Here it is worth mentioning some relevant studies on disinfectants tested against SARS-CoV-2 in vitro as well as in vivo, to reduce the viral shedding time from patients affected by mild-moderate or asymptomatic disease (PMID: 36676046; PMID: 36432693; PMID: 36432693)

Answer: We would like to thank reviewer 1 for the useful reference PMID: 36432693) and added the following suggestion to the conclusion: “…treating the underlying cause of nose picking (e.g. by using saline spray to reduce mucus) or using nasal disinfectant spray in SARS-CoV-2 infected individuals to counteract viral shredding.” (Conclusion, page 9, line 272-273) 

Reviewer #2: General comments

This interesting study investigated the association between nose-picking, nail-biting, glass-wearing and beard-keeping and COVID-19 infection among healthcare workers in the Netherlands.

The study is well-conceived and conducted with sound methodology and appropriate statistical methods.

The results are presented clearly and subsequent discussion was in line with the study outcome.

Limitations were also well-highlighted.

Specific comments

Table 1 to be re-formatted to include the lateral border

Answer: Table 1 has been re-formatted so it fits the page.

---

## [Editor Report · Decision Letter 1]

26 Jun 2023

Why not to pick your nose: association between nose picking and SARS-CoV-2 incidence, a cohort study in hospital health care workers

PONE-D-23-03063R1

Dear Dr. Lavell,

We’re pleased to inform you that your manuscript has been judged scientifically suitable for publication and will be formally accepted for publication once it meets all outstanding technical requirements.

Kind regards,

Tope Michael Ipinnimo

Academic Editor

PLOS ONE

---

## [Editor Report · Acceptance letter]

13 Jul 2023

PONE-D-23-03063R1 

Why not to pick your nose: association between nose picking and SARS-CoV-2 incidence, a cohort study in hospital health care workers 

Dear Dr. Lavell:

I'm pleased to inform you that your manuscript has been deemed suitable for publication in PLOS ONE. Congratulations! Your manuscript is now with our production department. 

Kind regards, 

on behalf of

Dr. Tope Michael Ipinnimo 

Academic Editor

PLOS ONE